# Transfer Learning with Neural AutoML

**Catherine Wong**
MIT
catwong@mit.edu

**Neil Houlsby**
Google Brain
neilhoulsby@google.com

**Yifeng Lu**
Google Brain
yifenglu@google.com

**Andrea Gesmundo**
Google Brain
agesmundo@google.com

## Abstract

We reduce the computational cost of Neural AutoML with transfer learning. AutoML relieves human effort by automating the design of ML algorithms. Neural AutoML has become popular for the design of deep learning architectures, however, this method has a high computation cost. To address this we propose Transfer Neural AutoML that uses knowledge from prior tasks to speed up network design. We extend RL-based architecture search methods to support parallel training on multiple tasks and then transfer the search strategy to new tasks. On language and image classification tasks, Transfer Neural AutoML reduces convergence time over single-task training by over an order of magnitude on many tasks.

## 1   Introduction

Automatic Machine Learning (AutoML) aims to find the best performing learning algorithms with minimal human intervention. Many AutoML methods exist, including random search [1], performance modelling [2, 3], Bayesian optimization [4], genetic algorithms [5, 6] and RL [7, 8]. We focus on neural AutoML, that uses deep RL to optimize architectures. These methods have shown promising results. For example, Neural Architecture Search has discovered novel networks that rival the best human-designed architectures on challenging image classification tasks [9, 10].

However, neural AutoML is expensive because it requires training many networks. This may require vast computations resources; Zoph and Le [8] report 800 concurrent GPUs to train on Cifar-10. Further, training needs to be repeated for every new task. Some methods have been proposed to address this cost, such as using a progressive search space [11], or by sharing weights among generated networks [12, 13]. We propose a complementary solution, applicable when one has multiple ML tasks to solve. Humans can tune networks based on knowledge gained from prior tasks. We aim to leverage the same information using transfer learning.

We exploit the fact that deep RL-based AutoML algorithms learn an explicit parameterization of the distribution over performant models. We present Transfer Neural AutoML, a method to accelerate network design on new tasks based on priors learned on previous tasks. To do this we design a network that performs neural AutoML on multiple tasks simultaneously. Our method for multitask neural AutoML learns both hyperparameter choices common to multiple tasks and specific choices for individual tasks. We then transfer this controller to new tasks and leverage the learned priors over performant models. We reduce the time to converge in both text and image domains by over an order of magnitude in most tasks. In our experiments we save 10s of CPU hours for every task that we transfer to.

## 2 Methods

### 2.1 Neural Architecture Search

Transfer Neural AutoML is based on Neural Architecture Search (NAS) [8]. NAS uses deep RL to generate models that maximize performance on a given task. The framework consists of two components: a controller model and child models.

The controller is an RNN that generates a sequence of discrete actions. Each action specifies a design choice; for example, if the child models are CNNs, these choices could include the filter heights, widths, and strides. The controller is an autoregressive model, like a language model: the action taken at each time step is fed into the RNN as input for the next time step. The recurrent state of the RNN maintains a history of the design choices taken so far. The use of an RNN allows dependencies between the design choices to be learned. The sequence of design choices define a child model that is trained and evaluated on the ML task at hand. The performance of the child network on the validation set is used as a reward to update the controller via a policy gradient algorithm.

### 2.2 Multitask Training

We propose Multitask Neural AutoML, that searches for model on multiple tasks simultaneously. It requires defining a generic search space that is shared across tasks. Many deep learning models require the same common design decisions, such as choice of network depth, learning rate, and number of training iterations. By defining a generic search space that contains common architecture and hyperparameter choices, the controller can generate a wide range of models applicable to many common problems. Multitask training allows the controller to learn a broadly applicable prior over the search space by observing shared behaviour across tasks. The proposed multitask controller has two key features: learned task representations, and advantage normalization.

**Learned task representations** The multitask AutoML controller characterizes the tasks by learning a unique embedding vector for each task. This task-embedding allows to condition model generation on the task ID. The task-embeddings are analogous to word-embeddings commonly used for NLP, where each word is associated to a trainable vector [14].

Figure 1 (left) shows the architecture of the multitask controller at each time step. The task embedding is fed into the RNN at every time step. In standard single-task training of NAS, only the embedding of the previous action is fed into the RNN. In multitask training, the task embedding is concatenated to the action embedding. We also add a skip connection across the RNN cell to ease the learning of action marginal distributions. The task embeddings are the only task-specific parameters. One embedding is assigned to each task; these are randomly initialized and trained jointly with the controller.

At each iteration of multitask training, a task is sampled at random. This task's embedding is fed to the controller, which generates a sequence of actions conditioned on this embedding. The child model defined by these actions is trained and evaluated on the task, and the reward is used to update the task-agnostic parameters and the corresponding task embedding.

**Task-specific advantage normalization** We train the controller using policy gradient. Each task defines a different performance metric which we use as reward. The reward affects the amplitude of the gradients applied to update the controller's policy, $\pi$. To maintain a balanced gradient updates across tasks, we ensure that the distribution of each task's rewards are scaled to have same mean and variance.

The mean of each task's reward distribution is centered on zero by subtracting the expected reward for the given task. The centered reward, or advantage, $A_\tau(m)$, of a model, $m$, applied to a task, $\tau$, is defined as the difference between the reward obtained by the model, $R_\tau(m)$, and the expected reward for the given task, $b_\tau = \mathbb{E}_{m \sim \pi}[R_\tau(m)]$: $A_\tau(m) = R_\tau(m) - b_\tau$. $b_\tau$. Subtracting such a baseline is a standard technique in policy gradient algorithms used to reduce the variance of the parameter updates [15].

The variance of each task's reward distribution is normalized by dividing the advantage by the standard deviation of the reward: $A'_\tau(m) = (R_\tau(m) - b_\tau)\sigma_\tau^{-1}$. Where $\sigma_\tau = \sqrt{\mathbb{E}_{m \sim \pi}[(R_\tau(m) - b_\tau)^2]}$. We

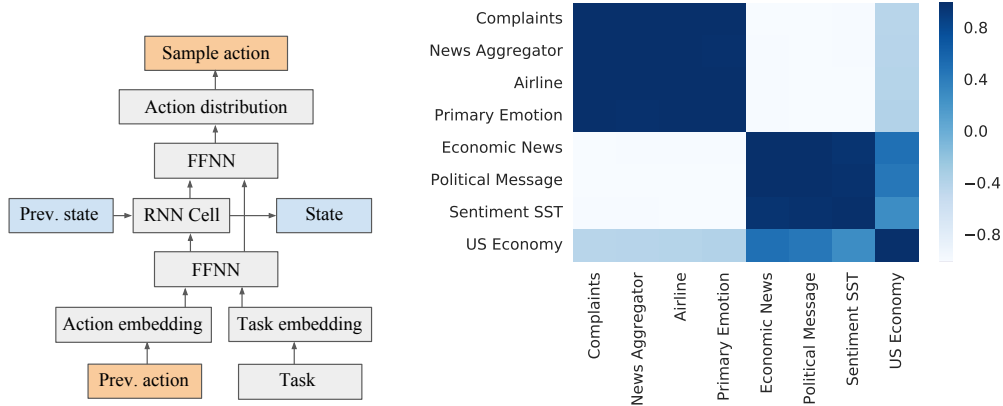

Figure 1: *Left:*A single time step of the recurrent multitask AutoML controller, in which a single action is taken. The task embedding is concatenated with the embedding of the action sampled at the previous timestep and passed into the controller RNN. All parameters, other than the task embeddings, are shared across tasks. *Right:* Cosine similarity between the task embeddings learned by the multitask neural AutoML model.

refer to $A'$ as the normalized advantage. The gradient update to the parameters of the policy $\theta$ is the product of the advantage and expected derivative of the log probability of sampling an action: $A'_\tau(m)\mathbb{E}_\pi[\nabla_\theta \log \pi_\theta(m)]$. Thus, normalizing the advantage may also be seen as adapting the learning rate for each task.

In practice, we compute $b_\tau$ and $\sigma_\tau$ using exponential moving averages over the sequence of rewards: $b_\tau^t = (1-\alpha)b_\tau^{t-1} + \alpha R_\tau(m)$, $\sigma_\tau^{2,t} = (1-\alpha)\sigma_\tau^{2,t-1} + \alpha(R_\tau(m) - b_\tau^t)^2$, where $t$ indexes the trial, and $\alpha = 0.01$ is the decay factor.

## 2.3 Transfer Learning

The multitask controller is pretrained on a set of tasks and learns a prior over generic architectural and parameter choices, along with task-specific decisions encoded in the task embeddings. Given a new task, we can perform transfer of the controller by: 1) reloading the parameters of the pretrained multitask controller, 2) adding a new randomly initialized task embedding for the new task. Then, architecture search is resumed, and the controller's parameters are updated jointly with the new task embedding. By learning an embedding for the new task, the controller learns a representation that biases towards actions that performed well on similar tasks.

## 3   Related Work

A variety of optimization methods have been proposed to search over architectures, hyperparameters, and learning algorithms. These include random search [1], parameter modeling [3], meta-learned hyperparameter initialization [16], deep-learning based tree searches over a predefined model-specification language [17], and learning of gradient descent optimizers [18, 19]. An emerging body of neuro-evolution research has adapted genetic algorithms for these complex optimization problems [20], including to set the parameters of existing deep networks [21], evolve image classifiers [5], and evolve generic deep neural networks [6].

Our work relates closest to NAS [8]. NAS was applied to construct CNNs for the CIFAR-10 image classification and RNNs for the Penn Treebank language modelling. Subsequent work reduces the computational cost for more challenging tasks [10]. To engineer an architecture for ImageNet classification, Zoph et al. [10] train the NAS controller on the simpler CIFAR-10 task and then transfer the child architecture to ImageNet by stacking it. However, they did not transfer the controller model itself, relying instead on the intuition that additional depth is necessary for the more challenging task. Other works apply RL to automate architecture generation and also reduce the computation cost. MetaQNN sequentially chooses CNN layers using Q-learning [22]. MetaQNN uses an aggressive exploration to reduce search time, though it can cause the resulting architectures to underperform.

Cai et al. [23] transform existing architectures incrementally to avoid generating entire networks from scratch. Liu et al. [11] reduce search time by progressively increasing architecture complexity, and [12] propose child-model weight sharing to reduce child training time.

Transfer learning has achieved excellent results as an initialization method for deep networks, including for models trained using RL [24, 25, 26]. Recent meta-learning research has broadened this concept to learn generalizable representations across classes of tasks [27, 28]. Simultaneous multitask training can facilitate learning between tasks with a common structure, though retaining knowledge effectively across tasks is still an active area of research [29, 30]. There is also prior research on transfer of optimizers for Neural AutoML; Sequential Model-based Optimizers have been transferred across tasks to improve hyperparameter tuning [31, 32], we propose a parallel solution for neural methods.

## 4 Experiments

**Child models**  Constructing the search space needs human input, so we choose wide parameter ranges to minimize injected domain expertise. Our search space for child models contains two-tower feedforward neural networks (FFNN), similar to the wide and deep models in Cheng et al. [33]. One tower is a deep FFNN, containing an input embedding module, fully connected layers and a softmax classification layer. This tower is regularized with an L2 loss. The other is a wide-shallow layer that directly connects the one-hot token encodings to the softmax classification layer with a linear projection. This tower is regularized with a sparse L1 loss. The wide layer allows the model to learn task-specific biases for each token directly. The deep FFNN's embedding modules are pretrained[1].This results in child models with higher quality and faster convergence.

The single search space for all tasks is defined by the following sequence of choices: 1) Pretrained embedding module. 2) Whether to fine-tune the embedding module. 3) Number of hidden layers (HL). 4) HL size. 5) HL activation function. 6) HL normalization scheme to use. 7) HL dropout rate. 8) Deep column learning rate. 9) Deep column regularization weight. 10) Wide layer learning rate. 11) Wide layer regularization weight. 12) Training steps. The Appendix contains the exact specification. The search space is much larger than the number of possible trials, containing 1.1B configurations. All models are trained using Proximal Adagrad with batch size 100. Notice that this search space aims to optimize jointly the architecture and hyperparameters. While standard NAS search spaces are defined strictly over architectural parameters.

**Controller models**  The controller is a 2-layer LSTM with 50 units. The action and task embeddings have size 25. The controller and embedding weights are initialized uniformly at random, yielding an approximate uniform initial distribution over actions. The learning rate is set to $10^{-4}$ and it receives gradient updates after every child completes. We tried four variants of policy gradient to train the controller: REINFORCE [34], TRPO [35], UREX [36] and PPO [37]. In preliminary experiments on four NLP tasks, we found REINFORCE and TRPO to perform best and selected REINFORCE for the following experiments.

We evaluate three controllers. First, Transfer Neural AutoML, our neural controller that transfers from multitask pre-training. Second, Single-task AutoML, which is trained from scratch on each task. Finally, a baseline, Random Search (RS), that selects action uniformly at random.

**Metrics**  To measure the ability of the different AutoML controllers to find good models, we compute the average accuracy of the topN (accuracy-topN) child models generated during the search. We select the best topN models according to accuracy on the validation set. We then report the validation and test performance of these models.

We assess convergence rates with two metrics: 1) accuracy-topN achieved with a fixed budget of trials, 2) the number of trials required to attain a certain reward. The latter can only be used with validation accuracy-topN since test accuracy-topN does not necessarily increase monotonically with the number of trials.

| Dataset | RS | NAML | T-NAML | | Dataset | RS | NAML | T-NAML |
|---|---|---|---|---|---|---|---|---|
| 20 Newsgroups | 2470 | 1870 | **435** | | 20 Newsgroups | 87.5 | 87.4 | **88.1±0.4** |
| Brown Corpus | 245 | 235 | **10** | | Brown Corpus | 37.0 | 38.2 | **53.4±3.3** |
| SMS Spam | 4815 | 3390 | **70** | | SMS Spam | 97.9 | 97.8 | **98.1±0.1** |
| Corp Messaging | 3850 | 1510 | **80** | | Corp Messaging | **90.0** | 90.2 | **90.2±0.3** |
| Disasters | 4970 | 2730 | **25** | | Disasters | 81.7 | 81.5 | **82.1±0.3** |
| Emotion | 4995 | 1645 | **195** | | Emotion | 33.9 | 33.7 | **35.3±0.3** |
| Global Warming | 4985 | 1935 | **90** | | Global Warming | 82.4 | **82.8** | **82.9±0.3** |
| Prog Opinion | 4200 | 3620 | **60** | | Prog Opinion | 68.9 | 66.3 | **70.3±0.9** |
| Customer Reviews | 4895 | 925 | **15** | | Customer Reviews | 77.8 | 79.0 | **81.4±0.5** |
| MPQA Opinion | 4965 | 1510 | **15** | | MPQA Opinion | 87.9 | 87.9 | **88.6±0.3** |
| Sentiment Cine | 4520 | 3225 | **535** | | Sentiment Cine | 73.2 | **76.3** | 75.4±0.4 |
| Sentiment IMDB | 4760 | **630** | 690 | | Sentiment IMDB | 85.8 | 87.3 | **88.1±0.1** |
| Subj Movie | 4745 | 1600 | **105** | | Subj Movie | 92.6 | 93.2 | **93.4±0.2** |

Table 1: Performance of Random Search (RS), single-task Neural AutoML (NAML) and Transfer Neural AutoML (T-NAML). Bolding indicates the best controller, or within ±2 s.e.m.. *Left*: Number of trials needed to attain a validation accuracy-top10 equal to the best achieved by Random Search with 5000 trials (250/2500 for Brown and 20 Newsgroups, respectively). *Right*: Test accuracy-top10 given at a fixed budget $B$ of 500 trials ($B = 250$ for Brown). Error bars show ±2 s.e.m. computed across the top 10 models. Similar s.e.m. values are observed for all methods.

## 4.1 Natural Language Processing

**Data**   We evaluate using 21 text classification tasks with varied statistics. The dataset sizes range from 500 to 420k datapoints. The number of classes range from 2 to 157, and the mean length of the texts, in characters, range from 19 to 20k. The Appendix contains full statistics and references.

Each child model is trained on the training set. The accuracy on the validation set is used as reward for the controller. The topN child models, selected on the validation set, are evaluated on the test set. Datasets without a pre-defined train/validation/test split, are split randomly 80/10/10.

The multitask controller is pretrained on 8 randomly sampled tasks: Airline, Complaints, Economic News, News Aggregator, Political Message, Primary Emotion, Sentiment SST, US Economy. We then transfer from this controller to each of the remaining 13 tasks.

**Results**   To assess the controllers' ability to optimize the reward (validation set accuracy) we compute the speed-up versus the baseline, RS. We first compute accuracy-top10 on the validation set for RS given a fixed budget of $B$ trials. We use $B = 5000$, except for the Brown Corpus and 20 Newsgroups where we can only use a $B = 500, 3500$, respectively, because these datasets were slower to train. We then report the number of trials required by AutoML and T-AutoML to achieve the same validation accuracy-top10 as RS with $B$ trials. Table 1 (left) shows the results. Note that RS may exhibit fewer than $B = 5000$ trials if it converged earlier. These results shows that T-AutoML is effective at optimizing validation accuracy, offering a large reduction in time to attain a fixed reward. In 12 of the 13 datasets T-AutoML achieves the desired reward fastest, and in 9 cases achieves an order of magnitude speed-up.

Next, we assess the quality of the models on the test set. Table 1 (right) shows test accuracy-top10 with a budget of 500 trials (250 for Brown Corpus). Within this budget, T-AutoML performs best on all but one dataset. T-AutoML outperforms single-task AutoML on 10 out of the 13 datasets, ties on one, and loses on two. On the datasets where T-AutoML does not produce the best final model at 500 trials, it often produces better models at earlier iterations. Figure 2 shows the full learning curves of test set accuracy-top10 versus number of trials. Figure 2 shows that in most cases the controller with transfer starts with a much better prior over good models. On some datasets the quality is improved with further training e.g. Emotion, Corp Messaging, but in others the initial configurations learned from the multitask model are not improved.

For reference, we put the learning curves for the initial multitask training phase in the Appendix. We also ran RS and single-task AutoML on these datasets. Slightly disappointingly, multitask training

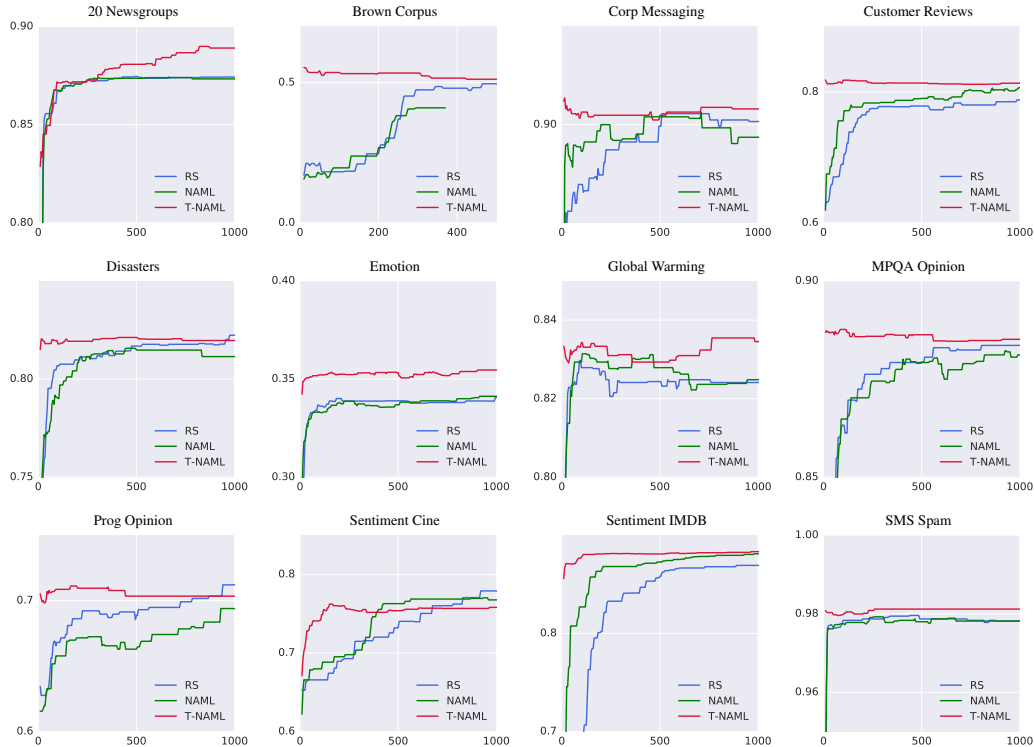

Figure 2: Learning curves for Random Search (RS), single-task Neural AutoML (NAML), and Transfer (T-NAML). *x-axis*: Number of trials (child model evaluations). *y-axis*: Average test set accuracy of the 10 models with best validation accuracy (test accuracy-top10) found up to each trial.

did not in itself yield substantial improvements over single-task; it attains a higher accuracy on two datasets, and in similar on the other six.

We aim to to attain good performance with fewest possible trials. We do not seek to beat state-of-the-art all datasets because first, although our search space is large, it does not contain all performant model components (e.g. convolutions). Second, we use embedding modules pretrained on large datasets which makes the results incomparable to those that only uses in-domain training data.

However, to confirm that Neural AutoML generates good models we compare to some previous published results where available. Overall we find that Transfer AutoML with the search space described above yields models competitive with the state-of-the-art. For example, Almeida et al. [38] use classical ML classifiers (Logistic Regression, SVMs, etc.) on SMS Spam and report best accuracy of 97.59%. Transfer AutoML gets accuracy-top10 of 98.1%. Le and Mikolov [39] report 92.58% accuracy on Sentiment IMDB with more complex architectures, Transfer AutoML's is a little behind, accuracy top-10 is 88.1%. Li et al. [40] report 86.8% accuracy using an ensemble of weighted neural BOWs on MPQA. Transfer AutoML achieve accuracy-top10 of 88.6%. Li et al. [40] also evaluate their ensemble of weighted neural BOW models on Customer Reviews, and achieve 82.5% best accuracy, though the best accuracy of any single model is 81.1%. Comparably, T-AutoML gets an accuracy-top10 of 81.4%. Barnes et al. [41] compare many algorithms and report best accuracy on Sentiment-SST of 83.1% using LSTMs. Multitask AutoML gets an accuracy-Top10 of 83.4%. The best performance achieved with a more complex architecture that is not in our search space is: 87.8% [39]. Maas et al. [42] report 88.1% on Movie Subj, Transfer AutoML gets accuracy-top10 of 93.4%.

**Computational Cost and Savings**  The median cost to perform a single trial across all 21 datasets in our experiments is $T = 268s$. If we run $B$ trials with a speedup factor of $S$, we save $BT(1 - S^{-1})/3600$CPU-h per task to attain a fixed reward (validation accuracy-top10). Estimating the speedup factors from Table 1 (left) for transfer over single-task, we attain a median computational saving of 30CPU-h per task when performing $B = 500$ trials. The mean is 89CPU-h, but this is heavily influenced by the slow Brown Corpus. The time to train the multitask controller is 15h on

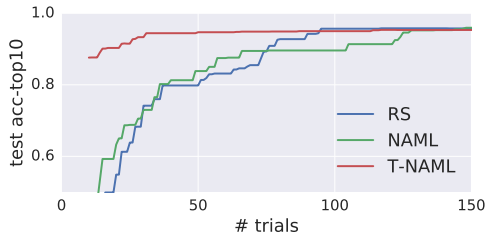

Figure 3: Comparison on an image classification task, Cifar-10. Mean test accuracy of the top 10 models chosen on the validation set.

100 CPUs. If we do not need the $M$ models for the tasks used to train the multitask controller, then we must run $> (1 - 1/S)^{-1}M$ new tasks to amortize this cost. For the median speedup in our experiments $S = 22$ that is $> 1.05M$ new tasks.

## 4.2 Image classification

To validate the generality of our approach we evaluate on image classification task: Cifar-10. We compare the same three controllers: RS, AutoML trained from scratch, and Transfer AutoML pretrained on MNIST and Flowers[2]. Figure 3 shows the mean accuracy-top-10 on the test set. The transferred controller attains an accuracy-top-10 of 96.5%, similar to the other methods, but converges much faster as in the NLP tasks. The best models embed images with a finetuned Inception v3 network, pretrained on ImageNet. Relu activations are preferred over Swish [43] and the dropout rate of converges to 0.3.

## 4.3 Analysis

**Meta overfitting**   The controller is trained on the tasks' validation sets. Overfitting of AutoML to the validation set is not often addressed. This type of overfitting may seem unlikely because each trial is expensive, and many trials may be required to overfit. However, we observe it in some cases.

Figure 4 (left, center) shows the accuracy-top10 on the validation and test sets on the Prog Opinion dataset. Transfer Neural AutoML attains good solutions in the first few trials, but afterwards its validation performance grows while test performance does not. The generalization gap between the validation and test accuracy increases over time. This is the most extreme case we observed, but some other datasets exhibit some generalization gap also (see Appendix for all validation curves). This effect is largest on Prog Opinion because the validation set is tiny, with only 116 examples.

Overfitting arises from bias due to selecting the best models on the validation set. Child evaluation contains randomness due to the stochastic training procedure. Therefore, over time we see an improved validation score, even after convergence, due to lucky evaluations. However, those apparent improvements are not reflected on the test set. Transfer AutoML exhibits more overfitting than single-task because it converges earlier. We confirmed this effect; if we 'cheat' and select models by their test-set performance, we observe the same artificial improvement on the test score as on the validation score. Other than entropy regularization, we do not combat overfitting extensively. Here, we simply emphasize that because our Transfer Neural AutoML model observes many trials in total, meta-overfitting becomes a bigger issue. We leave combatting this effect to future research.

**Distant transfer: across languages**   The more distant the tasks, the harder it is to perform transfer learning. The Sentiment Cine task is an outlier because it is the only Spanish task. Figure 2 and Table 1 show poorer performance of transfer on this task.

The most language-sensitive parameters are the pretrained word embeddings. The controller selects from eight pretrained embeddings (see Appendix), six of which are English, and two Spanish. In the first 1500 iterations, the transferred controller chooses English embeddings, limiting the performance. However, after further training, the controller switches to Spanish tables at around 2000th trial, Figure 4 (right). At trial 2000, T-AutoML attains a test accuracy-top10 of 79.8%, approximately equal to that or random search with 79.4%, and greater than single-task with 78.1%. This indicates that although transfer works best on similar tasks, the controller is still able to adapt to outliers given sufficient training time.

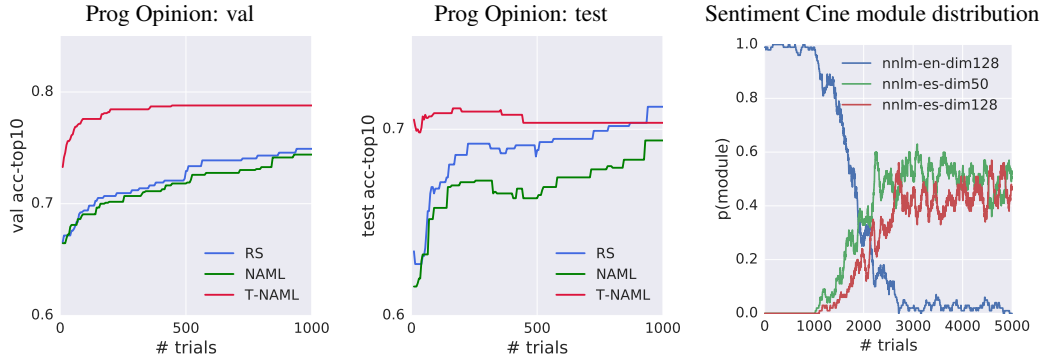

Figure 4: *Left, Center*: Learning curves on the validation (left) and test sets (center) for the Prog Opinion dataset. *Right*: Evolution of the choice of pretrained embedding module for transfer to the Spanish Corpus-Cine task. y-axis indicates the probability of sampling each table. This probability is estimated from the samples using a sliding window of width 100.

**Task representations and learned models** We inspect the learned task similarities via the embeddings. Figure 1 (right) shows the cosine similarity between the task embeddings learned during multitask training. The model assigns most tasks to two clusters. It is hard to guess *a priori* which tasks require similar models; the dataset sizes, number of classes and text lengths differ greatly. However, the controller assigns the same model to tasks within the same cluster. At convergence, the cluster {Complaints, New Agg, Airline, Primary Emotion} is assigned (with high probability) a 1-layer networks with 256 units, Swish activation function, wide-layer learning rate 0.01, and dropout rate 0.2. The cluster {Economic News, Political Emotion, Sentiment SST} is assigned 2-layer networks with 64 units, Relu activation, wide-layer learning rate 0.003, and dropout rate 0.3.

Other choices follow similar distributions for each cluster. For example, the same 128D word embeddings, trained using a Neural Language Model are chosen. The controller also always chooses to fine-tune these embeddings. The controller may remove either the deep or wide tower by setting the regularization very high, but in all cases it chooses to keep both active.

**Ablation** We consider two ablations of T-NAML. First, we remove the task embeddings. For this, we train a task-agnostic multitask controller without task embeddings, then transfer this controller as for T-NAML. Second, we transfer a single architecture rather than the controller parameters. For this, we train the task-agnostic multitask controller to convergence, and select the final child model. We then re-train this single architecture on each new task. Omitting task embeddings performs well on some tasks, but poorly on those that require a model different to the mode. Overall, according to accuracy-top10 at 500 trials, T-NAML outperforms the version without task embeddings on 8 tasks, loses 4, and draws on 1. The mean performance drop when ablating task embeddings is 1.8%. Using just a single model performs very poorly on many tasks, T-NAML wins 8 cases, loses 2, and draws 3, with a mean performance increase of 4.8%.

## 5 Conclusion

Neural AutoML, whilst becoming popular, comes with a high computational cost. To address this we propose transfer learning of the controller and show a large reductions in convergence time across many datasets. Extensions to this work include: Broadening the search space to contain more models classes. Attempting transfer across modalities; some priors over hyperparameter combinations learned on NLP tasks may be useful for images or other domains. Making the controller more robust to evaluation noise, and addressing the potential to meta overfit on small datasets.

**Acknowledgments**

We are very grateful to Quentin de Laroussilhe, Andrey Khorlin, Quoc Le, Sylvain Gelly, the Tensorflow Hub team and the Google Brain team Zurich for developing software frameworks and many useful discussions.

## Footnotes

[1]The pretrained modules are distributed via TensorFlow Hub: `https://www.tensorflow.org/hub` .

[2]goo.gl/tpzfR1

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
