[Supplementary Material]

# Supplementary Material for Transfer Learning with Neural AutoML

Catherine Wong
MIT
catwong@mit.edu

Neil Houlsby
Google Brain
neilhoulsby@google.com

Yifeng Lu
Google Brain
yifenglu@google.com

Andrea Gesmundo
Google Brain
agesmundo@google.com

This document contains a description of the search space used in our experiments (Table 1), details of the pretrained modules for embedding text and images (Tables 2 and 3), and statistics for the datasets used (Table 4 and 5). It also contains the learning curves for Transfer Neural AutoML (Figures 1 and 2) and Multitask Neural AutoML (Figures 3 and 4) on the validation and test sets.

Table 1: The search space for our AML models.

| Parameter | Search Space |
| --- | --- |
| 1) Input embedding modules | Text input: refer to Table 2. |
| | Image input: refer to Table 3. |
| 2) Fine-tune input embedding module | {True, False} |
| 3) Number of hidden layers | {1, 2, 3, 5, 7} |
| 4) Hidden layers size | {8, 16, 32, 64, 128, 256} |
| 5) Hidden layers activation | {relu, swish} |
| 6) Hidden layers normalization | {none, batch norm, layer norm} |
| 7) Hidden layers dropout rate | {0.0, 0.01, 0.05, 0.1, 0.2, 0.3, 0.4, 0.5, 0.6} |
| 8) Deep tower learning rate | {0.001, 0.003, 0.01, 0.03, 0.1, 0.3, 1.0, 3.0} |
| 9) Deep tower regularization weight | {0.0, 0.00001, 0.0001, 0.001, 0.01, 0.1, disable deep tower} |
| 10) Wide tower learning rate | {0.001, 0.003, 0.01, 0.03, 0.1, 0.3, 1.0, 3.0} |
| 11) Wide tower regularization weight | {0.0, 0.00001, 0.0001, 0.001, 0.01, 0.1, disable wide tower} |
| 12) Number of training samples | {1000, 3000, 10000, 30000, 100000, 300000, 1000000} |

Table 2: Options for text input embedding modules. These are pre-trained text embedding tables, trained on datasets with different languages and size. The text input to these modules is tokenized according to the module dictionary and normalized by lower-casing and stripping rare characters. The embeddings of each token are aggregated with a mean BOW approach. We provide the handle for the modules that are publicly distributed via the TensorFlow Hub service (`https://www.tensorflow.org/hub`).

| Language/ID | Dataset size (tokens) | Embed dim. | Vocab. size | Training algorithm | TensorFlow Hub Handles Prefix: `https://tfhub.dev/google/` |
|---|---|---|---|---|---|
| Spanish-small | 50B | 50 | 995k | Lang. model | `nnlm-es-dim50-with-normalization/1` |
| Spanish-big | 50B | 128 | 995k | Lang. model | `nnlm-es-dim128-with-normalization/1` |
| English-small | 7B | 50 | 982k | Lang. model | `nnlm-en-dim50-with-normalization/1` |
| English-big | 200B | 128 | 999k | Lang. model | `nnlm-en-dim128-with-normalization/1` |
| English-wiki-small | 4B | 250 | 1M | Skipgram | `Wiki-words-250-with-normalization/1` |
| English-wiki-big | 4B | 500 | 1M | Skipgram | `Wiki-words-500-with-normalization/1` |
| English-news-small | 90B | 100 | 5.9M | CBOW | |
| English-news-big | 90B | 500 | 5.9M | CBOW | |

Table 3: Options for image input embedding modules. To map an image, the controller can choose among state of the art architectures pre-trained on ImageNet. The module consists in the pre-trained model up to the final layer of logits. We provide the handle for the modules that are publicly distributed via the TensorFlow Hub service (`https://www.tensorflow.org/hub`).

| Architecture | Dataset | Reference | TensorFlow Hub Handles Prefix: `https://tfhub.dev/google/` |
|---|---|---|---|
| MobileNet v1 | Imagenet | (Howard et al., 2017) | `imagenet/mobilenet_v1_100_224/feature_vector/1` |
| Inception v2 | Imagenet | (Ioffe & Szegedy, 2015) | `imagenet/inception_v2/feature_vector/1` |
| Inception v3 | Imagenet | (Szegedy et al., 2015) | `imagenet/inception_v3/feature_vector/1` |
| Resnet v1.101 | Imagenet | (He et al., 2015) | `imagenet/resnet_v1_101/feature_vector/1` |
| Resnet v1.50 | Imagenet | (He et al., 2015) | `imagenet/resnet_v1_50/feature_vector/1` |

Table 4: Statistics and references for the NLP classification tasks.

| Dataset | Train samples | Valid. samples | Test samples | Classes | Lang | Len (chars) | Reference |
|---|---|---|---|---|---|---|---|
| 20 Newsgroups | 15,076 | 1,885 | 1,885 | 20 | En | 2,000 | (Lang, 1995) |
| Airline | 11,712 | 1,464 | 1,464 | 3 | En | 104 | crowdflower.com |
| Brown Corpus | 400 | 50 | 50 | 15 | En | 20,000 | (Francis & Kuera, 1982) |
| Complaints | 146,667 | 18,333 | 18,334 | 157 | En | 1,000 | catalog.data.gov |
| Corp Messaging | 2,494 | 312 | 312 | 4 | En | 121 | crowdflower.com |
| Customer Reviews | 3,044 | 378 | 378 | 2 | En | 100 | (Hu & Liu, 2004) |
| Disasters | 8,688 | 1,086 | 1,086 | 2 | En | 101 | crowdflower.com |
| Economic News | 6,392 | 799 | 800 | 2 | En | 1,400 | crowdflower.com |
| Emotion | 32,000 | 4,000 | 4,000 | 13 | En | 73 | crowdflower.com |
| Global Warming | 3,380 | 422 | 423 | 2 | En | 112 | crowdflower.com |
| MPQA Opinion | 8,547 | 1,025 | 1,034 | 2 | En | 19 | (Deng & Wiebe, 2015) |
| News Aggregator | 338,349 | 42,294 | 42,294 | 4 | En | 57 | (Lichman, 2013) |
| Political Message | 4,000 | 500 | 500 | 9 | En | 205 | crowdflower.com |
| Primary Emotions | 2,019 | 252 | 253 | 18 | En | 87 | crowdflower.com |
| Prog Opinion | 927 | 116 | 116 | 3 | En | 102 | crowdflower.com |
| Sentiment Cine | 3119 | 382 | 377 | 2 | Spanish | 2,760 | (Cruz et al., 2008) |
| Sentiment IMDB | 19946 | 5054 | 25000 | 2 | En | 1,360 | (Maas et al., 2011) |
| Sentiment SST | 67,349 | 872 | 1,821 | 2 | En | 105 | (Socher et al., 2013) |
| SMS Spam | 4,459 | 557 | 557 | 2 | En | 81 | (Almeida et al., 2011) |
| Subj Movie | 8052 | 972 | 976 | 2 | En | 127 | (Pang et al., 2002) |
| US Economy | 3,961 | 495 | 495 | 2 | En | 305 | crowdflower.com |

Table 5: Statistics and references for the Image classification tasks.

| Dataset | Train samples | Valid. samples | Test samples | Classes | Image size | Reference |
|---|---|---|---|---|---|---|
| Cifar 10 | 45000 | 5000 | 10000 | 10 | 32x32x3 | (Krizhevsky et al.) |
| Mnist | 55000 | 5000 | 10000 | 10 | 28x28x1 | (LeCun & Cortes, 2010) |
| Flowers | 2018 | 552 | 550 | 5 | variablex3 | goo.gl/tpzfR1 |

Figure 1: Learning curves for transfer learning. X-axis depicts number of trials (T) performed for each task. Y-axis depicts the mean validation accuracy of the 10 models achieving top validation accuracy (validation accuracy-top10).

Figure 2: Learning curves for transfer learning. X-axis depicts number of trials (T) performed for each task. Y-axis depicts the mean test accuracy of the 10 models achieving top validation accuracy (test accuracy-top10).

Figure 3: Learning curves for multitask training. X-axis depicts number of trials (T) performed for each task. Y-axis depicts the mean validation accuracy of the 10 models achieving top validation accuracy (validation accuracy-top10).

Figure 4: Learning curves for multitask training. X-axis depicts number of trials (T) performed for each task. Y-axis depicts the mean test accuracy of the 10 models achieving top validation accuracy (test accuracy-top10).