[Reviews · NeurIPS 2018]

Reviewer 1



This paper applies both multi-task training and transfer learning to AutoML. The paper extends the ideas presented in the Neural Architectura Search (NAS) technique (Barret Zoph and Quoc V. Le. Neural architecture search with reinforcement learning. In ICLR, 2017). The authors maintain the two-layer solution, with one network "the controller" choosing the architectural parameters for the "child" network which is used to solve the targeted task. The performance of the child network is fed back to the controller network to influence its results. The novelty of this paper is in the way this two-layer solution is used. Instead of using the controller to decide the network for one task, multiple tasks are considered at once during the search phase (multi-task training). In addition, once the network is trained for a series of tasks, transfer learning is used to train the controller network to guide the design of the child network. With this technique in place, the accuracy performance of the child network achieves close to state-of-the-art results in a lot fewer iterations/trails in the search space, considerably reducing the time it takes to find the design of a performant network. The results of the paper discuss two different types of tasks: NLP text classification and image classification. The NLP side of the analysis is a lot richer than the image classification. There are 21 tasks that are split randomly into two groups: 8 tasks used for the multi-task pre-training, while the rest of the tasks are used in the transfer learning setting. The 8-task pre-trained model is transferred to each of the remaining tasks individually. The results show that the number of iterations for finding the top-10 best models is drastically reduced even when doing multi-task transfer learning for the neural architecture search. The performance of the top-10 models is on-par with state of the art numbers achieved by manually-designed network, which is a really nice result. There are several way in which the paper could improve. I will list them in the order of importance: 1. My main concern with the paper is in the discussion of the time cost and savings achieved by this technique. I would have liked to see a discussion on how long it takes to pre-train the network on the 8 tasks before actually transfering it to a new task. I understand that this time is amortized across training networks for a lot of tasks, but I think it is an important detail of the technique that shouldn't be ignored. 2. The image classification part of the paper, while it shows that the technique works for a different domain, hence hopefully it generalizes, is rather weak. I'm guessing the results are not as interesting also due to the smaller dataset and simpler task. 3. There are few parts in the paper that could be explained a bit better. In particular, it's not entirely clear to me what the NAML approach really is. I'm guessing in the NAML case, instead of a random search, the two-layer controller-child network is used with feedback for only one task being fed from the child to the controller. Is NAML really the approach in NAS? If not, what are the differences? Nit-picks: The arrangement of the figures is a bit strange. In particular, there doesn't seem to be a relationship among the graphs in Figure 4. I would try to rearrange them without prioritizing on esthetics, but rather on flow and understanding. Same goes for Figure 1. which is discussed towards the end of the paper. Overall, I find the main idea of the paper simple and neat and I enjoyed reading the paper. -------------------------- AFTER REBUTTAL: I really like the idea of applying Transfer Learning to AutoML. I think both areas are hot right now and having one example of transfer learning applied to AutoML is valuable to the community. That being said, for this paper to be referred to in the future, it needs to have clear writing for the Experimental Section. Most of the questions that were posed in the reviews stem from not understanding the experimental section and that's a pity. I hope the authors will take the feedback from these reviews (including the clarifications provided in the author response) and include them in the paper.

Reviewer 2



The paper proposed a multitask approach to Automatic Machine Learning (AML) using reinforcement learning and Neural Architecture Search (NAS) to reduce the search space for the hyper-parameter optimization and transfers the knowledge from the previous tasks to speed up the network design for the novel tasks. Zoph et. Al., ICLR 2017 performed transfer learning experiments by reusing the cell on a different task, although they considered character language modeling on the same dataset but the proposed approach used different datasets and achieved the transfer through the controller. The paper is well-written and easy to understand. The proposed transfer AML/NAS is interesting and have practical importance but lacks some important details such as task embeddings, baselines etc and further analysis with additional baselines will help the paper. It is unclear how the task representations are learned at each iteration. The paper mentioned that the task embeddings are generated similar to word embedding but it doesn't seems straight-forward from the details given in the paper. Since the network configurations are generated conditioned on the task representations (in addition to the action embedding), it will be helpful if the author(s) give some details on this part. The proposed approach shows there exist two clusters of tasks (among the 8 tasks in the multitask training) not only based on the task embeddings but also on the hyper-parameters. It will be helpful if you include the hyper-parameters chosen for the transfer tasks to see how the configurations are chosen based on the clusters. One of the puzzling questions is that why the multitask training (Figure 4 in the Appendix) did not achieve better results than the Single-Task NAS compared to the transfer training. In order of the transfer learning to give better performance, we expect that we get similar behavior in the multitask training. This is important for understanding how the transfer NAS outperforms other baselines considered in this paper. It is unclear how the baselines are chosen for the experiments. No details are given about the baselines in the Experiments. Does RS optimization apply for each task separately? Does the Single-task NAS learn separate controller for each task? Does Single-task NAS use task embeddings? The baselines considered in this paper are insufficient to understand the behaviors of the Transfer NAS. In addition to the baselines in the paper, I think the following baselines can give further insights into the above surprising behaviour: 1) Choose separate controller for each task (w/o task embedding) 2) one controller for all the tasks (w/o task embedding) 3) Use same fixed/RS hyper-parameter for all the tasks.

Reviewer 3



This paper proposes to decrease the computational cost of neural architecture search by applying transfer learning. Weight sharing has been previously considered for neural architecture search within the child models, but it seems that this work is the first to apply transfer learning in the controller to this setting. This is apparently a novel idea in the Deep RL based controller setting of neural architecture search although it has been done in other settings, for example, with matrix factorization. Overall the weakness of this paper to me is that the approach is of limited novelty and is very straightforward from past work on transfer learning and neural architecture search. On the other hand, the use case is compelling and I find the evaluation of the model to be relatively convincing, thorough, and well explained. That being said, it would be nice to also see experimental justification for some of the design choices like task embeddings and task specific advantage normalization.